# Perfluorocarbon Nanodroplets as Potential Nanocarriers for Brain Delivery Assisted by Focused Ultrasound-Mediated Blood–Brain Barrier Disruption

**DOI:** 10.3390/pharmaceutics14071498

**Published:** 2022-07-19

**Authors:** Charlotte Bérard, Stéphane Desgranges, Noé Dumas, Anthony Novell, Benoit Larrat, Mourad Hamimed, Nicolas Taulier, Marie-Anne Estève, Florian Correard, Christiane Contino-Pépin

**Affiliations:** 1Aix Marseille Univ, APHM, CNRS, INP, Inst Neurophysiopathol, Hôpital Timone, Service Pharmacie, 13005 Marseille, France; charlotte.berard@ap-hm.fr (C.B.); florian.correard@univ-amu.fr (F.C.); 2Avignon Université, Unité Propre de Recherche et d’Innovation, Équipe Systèmes Amphiphiles Bioactifs et Formulations Eco-Compatibles, 84000 Avignon, France; stephane.desgranges@univ-avignon.fr; 3Aix Marseille Univ, CNRS, INP, Inst Neurophysiopathol, 13005 Marseille, France; noe.dumas@gmail.com; 4Université Paris-Saclay, CEA, CNRS, Inserm, BioMaps, Service Hospitalier Frédéric Joliot, 91401 Orsay, France; anthony.novell@universite-paris-saclay.fr; 5Université Paris Saclay, CEA, CNRS, NeuroSpin/BAOBAB, 91191 Gif-sur-Yvette, France; benoit.larrat@cea.fr; 6Aix Marseille Univ, CNRS, INSERM, Institut Paoli-Calmettes, CRCM, SMARTc Unit, COMPO Inria—Inserm Project Team, 13005 Marseille, France; mourad.hamimed@univ-amu.fr; 7Sorbonne Université, CNRS, INSERM, Laboratoire d’Imagerie Biomédicale—LIB, 75006 Paris, France; nicolas.taulier@sorbonne-universite.fr

**Keywords:** nanoemulsions, droplets, brain diseases, blood–brain barrier, ultrasound, drug carrier

## Abstract

The management of brain diseases remains a challenge, particularly because of the difficulty for drugs to cross the blood–brain barrier. Among strategies developed to improve drug delivery, nano-sized emulsions (i.e., nanoemulsions), employed as nanocarriers, have been described. Moreover, focused ultrasound-mediated blood–brain barrier disruption using microbubbles is an attractive method to overcome this barrier, showing promising results in clinical trials. Therefore, nanoemulsions combined with this technology represent a real opportunity to bypass the constraints imposed by the blood–brain barrier and improve the treatment of brain diseases. In this work, a stable freeze-dried emulsion of perfluorooctyl bromide nanodroplets stabilized with home-made fluorinated surfactants able to carry hydrophobic agents is developed. This formulation is biocompatible and droplets composing the emulsion are internalized in multiple cell lines. After intravenous administration in mice, droplets are eliminated from the bloodstream in 24 h (blood half-life (t_1/2_) = 3.11 h) and no long-term toxicity is expected since they are completely excreted from mice’ bodies after 72 h. In addition, intracerebral accumulation of tagged droplets is safely and significantly increased after focused ultrasound-mediated blood–brain barrier disruption. Thus, the proposed nanoemulsion appears as a promising nanocarrier for a successful focused ultrasound-mediated brain delivery of hydrophobic agents.

## 1. Introduction

Central nervous system (CNS) disorders, such as neurodegenerative diseases and brain tumors (e.g., glioblastoma), are among the most difficult diseases to treat due to limited therapeutic options. Presently, in neurodegenerative diseases, such as Alzheimer’s or Parkinson’s disease, while approved drugs mainly alleviate symptoms temporarily without significantly altering disease progression, no new drug offers hope for a cure [1,2]. Regarding glioblastoma, which is the most common and aggressive primary brain tumor, the 5-year survival rate remains below 10% and therapeutic options are nearly unchanged over the last several years, despite recent therapeutic advances in the field of oncology [3,4]. One of the major obstacles to efficient intracerebral drug delivery is the presence of the blood–brain barrier (BBB). The BBB is composed of tight junctions between endothelial cells in the vascular endothelium as well as active transporters that restrict exchanges. It acts as a selective physical barrier for maintaining brain homeostasis, but unfortunately also prevents several therapeutic agents, including many anticancer drugs, from reaching the brain [5,6].

To overcome the BBB and improve the bioavailability of therapeutic agents in brain tissue, several strategies have been elaborated, comprising invasive and non-invasive technologies [7]. Among non-invasive technologies, biocompatible nanoparticles have been developed as drug carriers (e.g., nanocarriers (NCs)). NCs provide a multipurpose platform for loading a wide range of drugs, including poorly water-soluble drugs, thus, increasing their bioavailability and protecting them from premature degradation. NCs can also improve drug pharmacokinetics, tissue distribution, drug targeting to specific pathological sites and “on-demand” drug delivery through the design of stimuli-responsive systems [8]. However, developing an efficient drug delivery nanoplatform requires considering several features influencing BBB penetration and systemic delivery [7,9]. In addition, despite the promising advances made in preclinical animal models, the clinical translation of nanotherapeutics for brain diseases remains slow and often fails. Indeed, a rapid and successful clinical translation depends on meticulous NC characterization during its development and certain criteria must be perfectly studied and defined (i.e., physicochemical characteristics, biocompatibility, accumulation in cancer cells, pharmacokinetics and biodistribution) [10,11]. 

Although NCs have the potential to improve drug delivery, the BBB still imposes real challenges in achieving an efficient drug concentration into the brain. It is now well-established that the combination of gas-filled bubbles, also known as microbubbles (MB) (e.g., SonoVue^®^, Optison^®^, Definity^®^), and focused ultrasound (FUS) may be used to permeate the BBB locally, temporary, reversibly and safely in animals and humans [12,13,14,15,16]. FUS-mediated BBB disruption has been found to enhance the delivery into the brain of a wide range of free therapeutic agents (e.g., anticancer drugs, antibodies) as well as NCs (e.g., liposomes) across the BBB and the blood–tumor barrier [17,18,19]. Increasing NC delivery to the brain by FUS-mediated BBB opening depends on their physicochemical properties and, in particular, on their hydrodynamic diameter (Dh). Indeed, although the optimal Dh of NCs is subject to discussion, it is currently accepted that NCs should ideally be smaller than 100 nm and that the BBB closure dynamics guarantee a probability of intracerebral penetration that decays quickly as the Dh of NCs increases [20,21].

Over the last two decades, perfluorocarbon (PFC) nanodroplets have gained significant interest as ultrasound-responsive NCs [22,23]. PFC nanodroplets exhibit an improved stability in the blood circulation and can extravasate in tissues due to their nanometric size. Their liquid core makes them detectable by fluorine-19 magnetic resonance imaging (^19^F-MRI) [24]. PFC-nanoemulsions, made of PFC nanodroplets stabilized in water by a shell of variable composition, have been widely developed as theranostic agents, as they can be used for both therapy and imaging. However, the dual hydrophobicity and lipophobicity of PFCs hinders solubilization of any chemical compound that is not fluorinated enough. To overcome this issue, one approach consists of adding a secondary phase inside the droplets, such as a hydrophobic oil compartment, able to dissolve the compound of interest.

For the present study, perfluorooctyl bromide (PFOB) was chosen to constitute the core of the nanodroplets, while in-house fluorinated surfactants called F-TACs were used to ensure the droplet stabilization and dispersion in water. PFOB stands out among PFCs for medical use as it exhibits low toxicity and an acceptable excretion profile (organ retention half-life of 3–4 days in humans) [25]. According to their fluorinated nature, F-TACs exhibit a high affinity for PFCs and are, thus, well suited to stabilize the core of PFC droplets. Moreover, they are highly water soluble (more than 150 g L^−1^ for F_6_TAC_7_), which makes them suitable candidates to disperse PFOB droplets in water [26]. F-TACs display a long half-life (30–50 h) and a good biocompatibility profile, showing no hemolytic activity at concentrations up to 200 g L^−1^ and display a high LD50 of 4.5 g kg^−1^ in rats after intravenous (i.v.) administration [27,28]. 

In this context, we developed F-TAC-stabilized PFOB emulsions, whose ripening can be overcome by a freeze-drying (FD) step following the emulsification process. To fulfill their therapeutic function, a biocompatible oil was used to encapsulate a hydrophobic molecule of interest (drug or dye) within the droplet core. Herein, we evaluated these droplets as suitable NCs for hydrophobic drugs by assessing their in vitro and in vivo behavior. Finally, we demonstrated, on healthy mice, their potential application as brain delivery systems when combined with an FUS-mediated BBB disruption technology using SonoVue^®^ MB (Figure 1).

## 2. Materials and Methods

### 2.1. Materials 

For nanodroplet preparation, 2,2′-azobisisobutyronitrile (AIBN), 98%, recrystallized from ethanol prior use, and Tributyl *O*-acetyl citrate (ATBC) 98% were purchased from Sigma-Aldrich (St. Quentin Fallavier, France), Capryol^®^ 90 (C-90) from GatteFossé SAS (Saint Priest, France), PFOB 98% from Fluorochem (Hadfiel, UK), and 1H,1H,2H,2H-perfluorohexanethiol and 1H,1H,2H,2H-perfluorooctanethiol were graciously provided by Atochem (Colombes, France). Tris(hydroxymethyl)acrylamidomethane (THAM) was synthesized as previously described by Pucci et al. [29]. Trehalose anhydrous (98%) was purchased from TCI Europe (Boereveldseweg, Belgium). 1,4-diazabicyclooctane (DABCO) 97% and DiIC18(5) or 1,1′-dioctadecyl-3,3,3′,3′-tetramethylindodicarbocyanine, 4-chlorobenzenesulfonate salt (DiD) were obtained from VWR (Strasbourg, France), and fluorescein isothiocyanate (FITC), isomer 1 95% from AlfaAeser (Karlsruhe, Germany). For in vitro studies, Dulbecco’s Modified Eagle Medium (DMEM), MCDB-131 culture medium, L-Glutamine, penicillin/streptomycin, trypsin (0.05% Trypsin/Ethylene diamine tetraacetic acid (EDTA) 1X) and Phosphate Buffer Saline (PBS) were purchased from Gibco (Paisley, UK). Eagle’s Minimum Essential Medium (EMEM) was obtained from Lonza (Verviers, Belgium) and fetal calf serum (FCS) from Eurobio (Les Ulis, France). Bovine Serum Albumin 98% (BSA) and paraformaldehyde (PFA) were purchased from Sigma-Aldrich (St. Quentin Fallavier, France). For analytic studies, HPLC Acetonitrile UltraHigh Performance Chromatography-Mass Spectrometry grade from Carlo-Erba (Val-De-Reuil, France) and all other reagents (sodium trifluoroacetate (TFA), trifluoracetic acid (99%)) and solvents were of reagent grade.

### 2.2. Synthesis of Fluorinated Surfactants 

All surfactants were easily synthesized by free radical polymerization in one step using two different perfluoroalkanethiols C_6_F_13_C_2_H_4_SH or C_8_F_17_C_2_H_4_SH (Fi) as transfer reagents (telogen) and AIBN as radical initiator. As such, 10 mL of solvent was used per gram of polymerizable monomer THAM (C = 0.57 M), and the concentration of AIBN was half that of telogen. R_0_ is the initial telogen/monomer molar ratio. The detailed synthesis of F-TAC surfactants is described elsewhere [29,30,31]. 

### 2.3. Synthesis of FITC-F_8_TAC_13_

F_8_TAC_13_ (100 mg, 1 eq, 0.036 mM) was dried under high vacuum for several hours and then dissolved in 8.9 mL of freshly distilled dry pyridine, followed by FITC (45.2 mg, 2.8 eq, 0.106 mM). DABCO was added until reaching a pH of 8. The mixture was heated at 50 °C for 48 h under a blanket of nitrogen. The mixture was precipitated in cold diethyl ether (Et_2_O, 225 mL) and centrifuged. The supernatant was removed and the pellet was washed 4 times with Et_2_O (20 mL). The precipitate was dried, dissolved in water and then freeze-dried to obtain a crude product (125 mg). The crude was dissolved in 3 mL of a mixture MeOH/H_2_O (9/1), filtered over 0.45 µm cellulose acetate (CA) filter and purified over LH2O in MeOH/H_2_O (9/1). Pure product was recovered (49 mg, yield = 42.2%). 

### 2.4. Preparation of “Oil-Free” Emulsions for Optimization Experiments

PFOB (0.2 mL) was added to a 50 mL centrifuge tube, followed by variable amounts of surfactant (F_6_TAC_6_, F_6_TAC_12_, F_6_TAC_29_, F_8_TAC_5_ or F_8_TAC_13_) dissolved in water (2 mL) with NaCl (0.9%). The mixture was cooled to 0 °C with an ice bath and insonified in continuous mode (CM) or pulsed mode (PM) (Duty Cycle (DC) 11.97%, pulse width 8 s) with a sonicator (VibraCell. TM. 75043, 750 W, Bioblock Scientific, Newtown, CT, USA), using a 13 mm sonotrode, situated at the bottom of the centrifuge tube, at various amplitudes from 20 to 80%. The sonication total duration varied from 1 to 3 min. The emulsion was centrifuged 1 min at 900× *g* before performing the size measurements. 

### 2.5. Preparation of “Baseline” Emulsions

To a dispersed phase (0.1 mL) (ATBC or C-90/PFOB mixture at various ratios (5%, 10%, 15% and 20%)), a surfactant aqueous solution (2 mL, 500 mg of F_8_TAC_13_ per mL of dispersed phase) was added, and the emulsification process was carried out in PM for 2 min (vide supra). The emulsion was centrifuged 1 min at 900× *g* before size measurement.

### 2.6. Preparation of Tagged “Baseline” Emulsions 

F_8_TAC_13_ (50 mg), for single-tagged emulsion (STE), or a mixture of F_8_TAC_13_-FITC (5 mg) and F_8_TAC_13_ (45 mg), for double-tagged emulsion (DTE), were dissolved in water (2 mL) with NaCl (0.9%). A fresh solution of hydrophobic dye DiD (0.5% *w*/*w*) in ATBC was prepared with the help of an ultrasound (US) bath and a vortex mixer. This dye-loaded oil (5 µL) was introduced in a 50 mL centrifuge tube, then PFOB (95 µL) was added. The surfactant aqueous solution was added, and the emulsification process was carried out as usual (vide supra). The obtained emulsion was centrifuged for 10 min at 5580× *g*. The supernatant was collected and centrifuged again at 25,830× *g* over 20 min to separate dye-loaded droplets (pellet) and unencapsulated dye (supernatant). Then the supernatant was discarded (1.5 mL) and the pellet rinsed twice with water (1.5 mL). The pellet was resuspended in water (1.5 mL) with the help of a vortex mixer. This workup was repeated once.

### 2.7. Preparation of the “Concentrated” Emulsion for Pharmacokinetics and Tissue Distribution Follow-Up 

Typically, ATBC oil (20 µL), PFOB (380 µL) and the aqueous surfactant solution (155 mg of F_8_TAC_13_ in 2 mL of water) were added to a 50 mL centrifuge tube. The mixture was insonified as usual (vide supra), centrifuged at 5800× *g* for 1 min and subsequently filtered over 0.22 µm CA filter.

### 2.8. Freeze-Drying (FD) Process 

Emulsions (baseline, tagged and concentrated) were prepared as previously described in dedicated sections. A stock solution of cryoprotectant (trehalose) in water was prepared ([C] = 357 mg mL^−1^). In 2 mL vials, an aliquot (300 µL) of emulsion was added along with various volumes of trehalose stock solution (to obtain final trehalose concentrations ranging from 5 to 20% *w*/*v*). After being frozen at −20 °C for 4 h, the vials were freeze-dried overnight. Each vial was kept at 4 °C until use. Each vial was reconstituted by adding the lost water extemporaneously before use.

### 2.9. Nanodroplet Characterization 

The particle size distribution data were obtained using a Dynamic Light Scattering (DLS) apparatus (Zeta Sizer Nano-S, Malvern Instruments, Orsay, France). Sample preparation was achieved by diluting the initial solution with deionized water by 10-times. These samples were placed in a 45 µL quartz tank and subjected to 6 measurements of 15 s each at 25 °C with a scattering angle of 173°. Refractive indexes assigned to PFOB and water (dispersant) were defined as 1.305 and 1.333, respectively. Aside from droplet optimization experiments, all DLS measurements were performed in triplicate. The hydrodynamic droplet mean diameter Dh is an average of 18 measurements and is reported as an intensity-based distribution with the cumulant method (Z-average). 

### 2.10. Transmission Electron Microscopy (TEM) Images 

TEM observations were conducted with a MET Hitachi 7800–120 kV instrument. A drop of emulsion was covered with a carbon-coated copper grid and allowed to equilibrate for 2 min. The grid was then stained with a 3% ammonium molybdate solution. Excess liquid was removed by gently touching the edge of grid with filter paper and the grid dried overnight at room temperature. After that, the grid was transferred to TEM operating at an acceleration voltage of 80 kV.

### 2.11. Determination of PFOB Volume Fraction in Emulsions by ^19^F-Nuclear Magnetic Resonance (^19^F-NMR) 

All spectra were recorded on a 400 MHz Bruker Avance II spectrometer with a resonance frequency of 376.53 MHz for ^19^F-NMR. Spectra were acquired using the inverse-gated decoupling technique. Each spectrum was the result of 256 scans with 131 072 data points using a relaxation delay of 4 s. Peak area was integrated using manufacturer standard software (Topspin, version 3.5pl7, Bruker, Wissembourg, France). All experiments were performed on triplicate samples. The detailed procedure is described elsewhere [30].

### 2.12. Oil and DiD Titration by HPLC

HPLC analysis was conducted on an Alliance Waters system (e2695) equipped with a 2998 photodiode array detector (Saint-Quentin, France). Titration was carried out with a Sunfire C18 Column from Waters (4.6 × 150 mm, 3.5 μm). A linear gradient of water (0.1% TFA) and acetonitrile (0.1% TFA) from 10% to 80% in 20 min was used. The detection wavelength was 214 nm for oil (ATBC and C90) titration and 227 nm for DiD titration.

### 2.13. Emulsion Stability Determination 

Freeze-dried baseline emulsions were resuspended with various volumes (150 or 300 µL) of 3 different dispersing phases (water, NaCl 0.9% or EMEM with 10% FCS) and were stored at 2 different temperatures (4 °C or 37 °C). The impact of dilution was also studied by diluting samples up to 1/100 after emulsion rehydration with 300 µL of water (storage at 4 °C) or EMEM with 10% FCS (storage at 37 °C). Dh was measured by DLS at 0, 1, 2, 4, 8 and 24 h after rehydration and/or dilution as described above.

For the study of DiD passive release, freeze-dried STE were resuspended with water (300 µL) and kept at 37 °C for either 0, 1, 2, 4, 8 or 24 h. At each time, emulsion was centrifugated for 1 h at 25,830 g and the supernatant was withdrawn (200 µL), diluted with water (completed to 600 µL) and titrated by HPLC. The DiD concentration of droplets at each time is the concentration of the entire emulsion at t = 0 minus that of each supernatant. Experiments were performed in triplicate. All points were normalized to the DiD concentration of droplets at t = 0.

### 2.14. Cell Culture 

Human glioblastoma cell line (U87-MG) and two non-cancerous cell lines (Human Microvascular Endothelial Cells (HMEC-1) and mouse astrocyte cells (C8-D1A)) were obtained from American Type Culture Collection (ATCC). U87-MG cells transfected with DsRed were kindly provided by Manon Carré [32]. HMEC-1 cells were cultured in MCDB-131 with 10% heat-inactivated FCS, 2 mM L-Glutamine, 1% (*v*/*v*) of penicillin/streptomycin and 10 ng mL^−1^ of epidermal growth factor. C8-D1A cells were maintained in DMEM with 10% FCS and 1% (*v*/*v*) of penicillin/streptomycin. U87-MG cells were maintained in EMEM with 10% FCS, 2 mM L-Glutamine and 1% (*v*/*v*) of penicillin/streptomycin. All cells were routinely maintained at 37 °C and in a 5% CO_2_ humidified incubator.

### 2.15. Biocompatibility Assay

Cells (9400 cells cm^−2^ for HMEC-1, 37,500 cells cm^−2^ for C8-D1A and 12,500 cells cm^−2^ for U87-MG) were seeded on 96-well plates and allowed to grow for 24 h (HMEC-1 and U87-MG cells) or 72 h (C8-D1A cells) before treatment. Then, cells were treated for 72 h with fresh medium containing various concentrations of different components of droplets: F_8_TAC_13_ (ranging from 0.05 to 10 g L^−1^), ATBC (0.01 to 100 mg L^−1^) and droplets themselves (0.046 to 4.6 g L^−1^ of F_8_TAC_13_, emulsion at 4.7% of Fv). Cell viability was assessed by using a colorimetric assay with 3-(4,5-dimethylthiazol-2-yl)-2,5-diphenyltetrazolium bromide (MTT) [33]. Viability was calculated as measurement of absorbance from a sample, normalized to the absorbance from control cells. 

### 2.16. Confocal Laser Scanning Microscopy (CLSM) 

U87-MG cells transfected with DsRed were seeded for imaging in an 8-well chambered cover glass (Lab-Tek) at a density of 10,000 cells per 300 µL per well and allowed to grow for 24 h prior to treatment. Then, cells were treated with fresh medium alone (control cells) or containing DTE (300 µL, emulsion at 2.6% of volume fraction (Fv), diluted to 1/50) for 2, 4, 8 or 24 h at 37 °C. The suspension was removed, cells were rinsed three times with PBS and fixed with a 4% PFA solution in PBS (4 °C, 10 min) then washed again with PBS or maintained in DMEM without phenol red + 10% FCS. For the cells that were fixed, cover glass was picked up and permanently mounted to the glass side using one drop of ProLong^TM^ antifade mountant (Invitrogen™, Paisley, UK). Cells were imaged using a Leica TCS SP5 CLSM (Leica Microsystems, Nanterre, France) equipped with an argon laser for green labeling (FITC, excitation 488 nm) and a helium-neon laser for red and blue labeling (DsRed, excitation 543 nm and DiD, excitation 633 nm, respectively). A Z-stack acquisition of 20 frames covering a depth of 12 µm was recorded for evaluation of the nanodroplet distribution across the section. Image analysis was performed using ImageJ software (NIH, Bethesda, MD, USA).

### 2.17. Fluorescence-Activated Cell Sorting (FACS) 

Cells (12,500 cells cm^−2^ for HMEC-1, 37,500 cells cm^−2^ for C8-D1A and 12,500 cellscm^−2^ for U87-MG) were seeded on 6-well plates. After 24 h (U87-MG and HMEC-1) or 72 h (C8-D1A) of growth at 37 °C, cells were exposed with fresh medium alone (control cells) or containing DTE (300 µL, emulsion at 2.6% of Fv, diluted to 1/50) for 1, 2, 4, 10 and 24 h. After exposure, the suspension was removed and cells were rinsed three times with PBS. After being detached with trypsin (300 µL), cells were washed then fixed for 10 min at room temperature with a mixture of 50% of 4% PFA solution and 50% of buffer solution containing PBS, 0.5% of BSA and 2 mM of EDTA and redispersed in the same buffer solution. Semi-quantitative studies were performed using flow cytometry (FACScalibur^TM^, BD Biosciences, Franklin Lakes, NJ, USA). A total of 10,000 events were counted per sample and data were recorded with CellQuest Pro Software (BD Biosciences, Franklin Lakes, NJ, USA) and analyzed using Flowing Software 2. The percentage of fluorescence doubly positive cells per 10,000 cells and the mean fluorescence intensity (MFI) of 10,000 treated cells were reported for each exposure time and were statistically (Student’s *t*-test) compared with the previous exposure time for each cell line.

Biocompatibility assay and FACS experiments were repeated three times independently and results were expressed as mean ± standard deviation.

### 2.18. In Vivo Studies 

In vivo whole blood pharmacokinetics, tissue distribution and intracerebral accumulation of droplets were determined using 6-week-old female C57BL/6 mice (Envigo, Indianapolis, IN, USA).

### 2.19. In Vivo Pharmacokinetics and Tissue Distribution 

Sixty-five mice were administered a single dose of concentrated emulsion (300 µL of emulsion, with mean PFOB Fv of samples at 13.2% (exact % assessed by ^19^F-NMR), i.e., a mean PFOB injection of 39.6 µL per mouse) by retro-orbital i.v. injection. To administer the nanoemulsion by retro-orbital injection in mice, a 30-gauge, 8 mm needle attached to a 0.5 mL insulin syringe was used (BD Micro-Fine^TM^, Becton, Dickinson and Co., Franklin Lakes, NJ, USA). Mice were divided into 13 groups (5 mice per group) corresponding to 13 selected time points for blood sampling (5, 15, 30, 60 min, 2, 4, 8, 16, 24, 48, 72 h, 7 and 14 days). At pre-determined time points, mice were anesthetized with an intraperitoneal (i.p.) injection of ketamine/xylazine (100 mg kg^−1^ and 10 mg kg^−1^, respectively, 0.1 mL per 10 g of weight) and sacrificed for blood collection. The whole blood was collected in heparin-coated tubes by cardiac puncture. For 7 pre-determined time points (1, 8, 24, 48, 72 h, 7 and 14 days) major organs (liver, spleen, kidneys, brain, lungs and pancreas) were also harvested after mice perfusion with PBS. Organs were weighed and homogenized in PBS (100 to 600 µL depending on the organs) with a T25 Ultra-Turrax^®^ (IKA, Staufen, Germany). Blood and organ samples were stored at −80 °C until determination of PFOB content by ^19^F-NMR. The whole blood and organ concentration–time profiles were obtained by averaging the dosage values of 5 mice per sampling time. To estimate pharmacokinetic parameters, non-compartmental analysis was performed using PKanalix software, version 2020R1 (Antony, France: Lixoft SAS, 2020). Maximum observed concentration (C_max_) and time to maximum observed concentration (T_max_) were determined directly from the concentration–time data. The area under the plasma concentration–time curve (AUC) and area under the first moment curve (AUMC) were calculated using the log-linear trapezoidal rule until the last sampling time with a quantifiable concentration (AUC_0–24h_ and AUMC_0–24h_) and extrapolated to infinity using the λz estimate (AUC_0-inf_ and AUMC_0-inf_), where λz is the terminal rate constant. The terminal half-life (t_1/2_) was subsequently calculated as ln2/λz. Estimates of clearance (Cl), mean residence time (MRT) and volume of distribution (Vd) were calculated as dose/AUC_0-inf_, AUMC_0-inf/_AUC_0-inf_ and Cl × MRT, respectively.

### 2.20. ^19^F-NMR PFOB Quantitative Determination in Mice Blood and Organs 

A concentrated emulsion was prepared and its exact Fv of PFOB was assessed by ^19^F-NMR as usual (vide supra). A range of dilutions was then performed with water to make the calibration curve as follows: murine blood (500 µL) was diluted with the former emulsion (100 µL) at various concentrations to reach a range of PFOB concentrations from 2.6 to 0.026%, then this solution (500 µL) was quantified by ^19^F-NMR. The PFOB titration of blood and organ samples (500 µL) was performed in the same manner, but by diluting the samples to analyze with water (100 µL) instead of with emulsion (100 µL).

### 2.21. In Vivo Hemispheric BBB Disruption Procedure 

Hemispheric BBB permeabilization was achieved using a focused transducer with a central frequency at 1.5 MHz (active diameter 25 mm, focal depth 20 mm, axial resolution 5 mm, lateral resolution 1 mm, Imasonic, France) connected to a single-channel programmable generator (Image Guided Therapy, Pessac, France). The transducer was mounted on a motorized XY*Z*-axis stage and positioned above the mouse head maintained under anesthesia. Mice were anesthetized with ketamine/xylazine cocktail (100 mg kg^−1^ and 10 mg kg^−1^, respectively, 0.1 mL per 10 g of weight, i.p.) and were placed in prone position. The transducer was coupled to the shaved head of mouse using a latex balloon filled with deionized and degassed water and coupling gel. A bolus (60 µL) of commercially available MB (SonoVue^®^, Bracco Imaging, Milan, Italy) was administered through the retro-orbital route. To administer SonoVue^®^ MB by retro-orbital injections in mice, a 1 mL tuberculin syringe (PentaFerte, Ferrara, Italy) with a 26-gauge, 12.5 mm needle (SOL-M^TM^, Sol-Millenium, Shanghai, China) was used. Then, ultrasound transcranial sonication (FUS) was started immediately after MB injection and was applied above the right hemisphere with an estimated in situ peak negative acoustic pressure of 0.33 MPa (0.45 MPa in free water) considering a mean transmission loss through the mouse skull of 25.7% [34]. Concretely, the FUS sequence was composed of ultrasonic waves transmitted for 5.6 s at 1.5 MHz with DC of 72%. A raster scan (X*Y*-axis, velocity: 10 mm s^−1^) was synchronized to the generator output in order to induce a hemispheric brain BBB opening of 6 mm (anterior-posterior) × 3.6 mm (lateral right hemisphere). The sequence was repeated 20 times for a total exposure of 112 s. Safe and efficient BBB disruptions in mice were demonstrated in previous studies using scanning patterns combined with high-DC US [35,36]. The left hemisphere was not sonicated and was used as control. After FUS, mice were moved to a clean cage and placed under a heating lamp until sacrifice. For BBB opening confirmation, one mouse was injected intravenously with Evans blue (50 µL of a 4% solution) after sonication. All mice had the same body weight.

### 2.22. Intracerebral Accumulation of STE Using Fluorescence Microscopy 

Ten minutes before FUS, 4 mice were injected i.v. (through retro-orbital route) with STE (300 µL). Mice were sacrificed 1 h post BBB permeabilization, perfused with saline through the left ventricle and fixed with 4% PFA solution. Brains were carefully collected and placed into 4% PFA solution. After 72 h of fixation at 4 °C, brains were cut into 50 µm thick slices using Vibratome (Leica Microsystems, Wetzlar, Germany) following sagittal axis. For each mouse, 42, 33, 34 and 30 coronal brain slices obtained could be analyzed. Images were obtained using a Zeiss AXIO-Observer Z1 microscope (Carl Zeiss, Oberkochen, Germany, 20×, excitation: 640/30 nm, emission: 690/50 nm, 250 ms exposure time, 25% overlap, 4 × 4 binning). To quantify fluorescence signals in the sonicated region, all brain slices were imaged using a Zeiss Axioscan microscope (Carl Zeiss, Oberkochen, Germany). Images were processed with Zen software (Carl Zeiss, Oberkochen, Germany). The sonicated and unsonicated areas were manually outlined, and the MFI of each area was quantified. To analyze intracerebral accumulation in the BBB-permeabilized hemisphere, MFI of sonicated hemisphere was compared to that of the contralateral control hemisphere.

### 2.23. Intracerebral Accumulation of STE Using FACS 

Ten minutes before FUS, 3 mice were injected i.v. (through retro-orbital route) with STE (300 µL). Mice were sacrificed 4 h post BBB disruption and perfused with NaCl 0.9% through the left ventricle. Brains were collected and the sonicated and unsonicated hemispheres were separated; 3 areas were collected in each hemisphere and dissociated with a mouse brain dissociation kit (Miltenyi Biotec, Bergisch Gladbach, Germany). Semi-quantitative analysis was performed using flow cytometry (FACScalibur^TM^, BD Biosciences, Franklin Lakes, NJ, USA). In total, 10,000 events were counted per each area of each hemisphere and data were recorded with CellQuest Pro Software (BD Biosciences, Franklin Lakes, NJ, USA) and analyzed using FlowJo software (v10, Mario Roederer, Ashland, OR, USA). The mean percentage of fluorescence-positive cells per 10,000 cells was calculated for each brain hemisphere and sonicated brain hemisphere was statistically compared to unsonicated (control) for each mouse (*n* = 3 areas of each hemisphere, Student’s *t*-test).

### 2.24. Histological Evaluation

Ten minutes before FUS, one mouse was injected intravenously (through retro-orbital route) with STE (300 µL) and one mouse was untreated. Twenty-four hours after hemispheric BBB disruption (as described above), mice were sacrificed, perfused with saline through the left ventricle and fixed with 4% PFA solution. Brains were carefully collected, fixed and conserved in formalin solution before paraffin-embedding. Four µm thick paraffin sections were then processed manually. Slides were deparaffinized in three successive baths of xylene (Hydroclear) for 10 min and then rehydrated with ethanol (100, 80, 70%). After rinsing in tap water for 10 min, slides were stained with Mayer’s Hematoxylin for 1 min. After washing, slides were stained with eosin for 15 s and rapidly rinsed in tap water. Slides were dehydrated using three baths of 100% ethanol followed by three baths in xylene and mounted with Eukitt medium (Sigma-Aldrich, St. Quentin Fallavier, France).

## 3. Results

### 3.1. Nanoemulsion Preparation and Characterization

#### 3.1.1. Surfactant Synthesis

FiTACn are in-house fluorinated surfactants that were already used to prepare nano- and micro-sized emulsions [30,31]. They consist of a non-ionic polar head comprising n repeated Tris(hydroxymethyl)aminomethane (TACn) units (*n* = DPn is the average degree of polymerization) and a hydrophobic perfluorinated tail (Fi), where i corresponds to the number of carbon atoms linked to fluor (Figure 1a). The DPn of the polar head (i.e., the length of the oligomeric backbone) depends on both the ratio of starting reactants and experimental conditions [37]. All surfactants were easily synthesized by free radical polymerization using two different perfluoroalkanethiols, C_6_F_13_C_2_H_4_SH or C_8_F_17_C_2_H_4_SH, as transfer reagents (telogen) and AIBN as radical initiator. Five surfactants were used for the nanoemulsion optimization, namely F_6_TAC_6_, F_6_TAC_12_, F_6_TAC_29_, F_8_TAC_5_ and F_8_TAC_13_; their detailed synthesis is described elsewhere [30].

#### 3.1.2. Nanoemulsion Synthesis and Droplet Dh Optimization 

In order to produce monodispersed nanoemulsions with a droplet size between 15 and 100 nm [21,38,39], a US tip was used at low frequency (20 kHz) among all the available methods, as it is a fast and convenient method. Several factors can influence the emulsion properties, such as the nature and concentration of surfactants, the nature of dispersed and continuous phases and the level of applied power [40,41]. These factors were investigated in the current study in order to reach adequate droplet Dh, measured by DLS.

The optimization of US parameters was performed with F_6_TAC_6_ surfactant, specifically insonification duration, amplitude and mode (CM vs. PM). Figure 1b,c show that the droplet Dh tends to be inversely proportional to the F_6_TAC_6_ concentration and US amplitude reaches a plateau at 250 mg mL^−1^ (Figure 1b) and 60% (i.e., 450 W) (Figure 1c), respectively. Moreover, the PM (DC of 11.97% and pulse width of 8 s) provides smaller droplets compared to CM and the optimum insonification time is found to be 2 min, since larger Dh and no size improvements were observed at 1 min and 3 min, respectively (Figure 1c). 

With the best insonification conditions ascertained, i.e., 60% amplitude (450 W) in PM over 2 min (DC 11.97%, pulse width 8 s), four other surfactants were screened at different concentrations. Figure 1d shows that for a given surfactant, increasing its concentration results in a Dh reduction until reaching a concentration threshold that leads to droplet growth. This effect, described in the literature as “emulsion over-processing”, can be ascribed to an excess of surfactant that decreases the rate of US wave diffusion and adsorption onto the newly formed droplet surface [42]. As shown in Figure 1, the smallest droplet size (68 ± 15 nm) was obtained with the F_8_TAC_13_ surfactant at a concentration of 500 mg mL^−1^ of dispersed phase (Φ_dispersed_), applying pulsed US (450 W) for 2 min, so these conditions (surfactant type, concentration and US parameters) were applied for the rest of the study.

#### 3.1.3. Oil introduction

To allow a substantial drug or dye solubilization in the core of PFC droplets, the introduction of an oil phase, partitioned with PFOB in the droplet core, is required, as only highly fluorinated compounds are soluble in PFC [43]. For this purpose, two biocompatible oils, namely ATBC and C-90, were tested at different oil/dispersed phase ratios. Both oils are Food-and-Drug-Administration-approved as pharmaceutical excipients and already used to solubilize hydrophobic drugs [44,45,46]. Interestingly, the introduction of this oily phase in the formulation resulted in an efficient decrease in droplet Dh (Table 1). For each experiment series, the ratio of oil in the dispersed phase was increased as long as the size distribution of droplets remained monodispersed (i.e., low polydispersity index (PDI), ideally PDI < 0.20).

As shown in Table 1, this limit was reached at 10% for ATBC and at 20% for C-90. In each case, once a bimodal size distribution appeared in DLS (PDI > 0.3), a filtration of the solution over a 0.22 µm CA filter allowed us to recover a monodispersed emulsion. At this limited oil concentration (10% for ATBC and 20% for C-90), the titration of both PFOB and oil phases by ^19^F-NMR and HPLC before and after filtration, respectively, indicates that the larger droplet population consists of oil only (Appendix A and Appendix A). Indeed, after filtration, no fluctuation in PFOB concentration is observed in the emulsion, while oil concentration decreases to 8.22% (*v*/*v*) and to 16.20% *v/v* for ATBC and C-90, respectively. Just below the limit oil concentration, for example, at 5% of ATBC, filtration over 0.22 µm CA enables one to reduce the PDI of the emulsion (from 0.296 to 0.167) (Table 1).

Both oils are good candidates to solubilize a hydrophobic agent within PFOB droplets of suitable diameter for biomedical applications. However, according to available in vivo safety data (ATBC, LD50 (i.p.) > 4 g kg^−1^ and C-90, maximum tolerated dose (i.v.) < 75 mg kg^−1^ in mice), ATBC (at 5% in the dispersed phase) was chosen as the oily phase for the whole study [47,48]. All emulsion samples prepared for biological evaluations were filtrated before use.

#### 3.1.4. Emulsion Freeze-Drying and Labeling 

As emulsions are thermodynamically unstable, the impact of FD to delay the emulsion ripening was investigated, to increase its shelf life while limiting physicochemical alteration. A preliminary assay demonstrated that the stress generated during the FD process leads to a destabilization of droplets (Appendix A). To overcome this problem, a cryoprotectant is usually required [49]. For this study, trehalose was chosen as cryoprotectant and used at various concentrations in emulsions with or without ATBC. As shown in the Appendix A), the best emulsion stabilization in terms of droplet size and PFOB composition was achieved with the lowest trehalose concentration (i.e., 5%) and when the dispersed phase contains 5% of ATBC. Aside from this, we can observe a decrease in cryoprotectant efficiency above a concentration of 20% *w*/*v*; this result is in accordance with the work of Li et al. [50]. Our dry formulation allows for the recovery, after the addition of water, of the parent nanoemulsion, with only a slight change in droplet size (around 60 nm) and composition, which could remove one of the barriers that hampers its development for clinical use [51]. To our knowledge, there are only a few examples of dry nanoemulsions developed for medicinal applications described in the literature [50,52,53,54,55]. Finally, the optimized emulsion consists of a PFOB/ATBC (95/5% *v*/*v*) dispersed phase and a water dispersing phase containing F_8_TAC_13_ (at a concentration of 500 mg mL^−1^ of dispersed phase) and trehalose (5% *w*/*v*). This formulation (estimated Fv = 4.7% according to emulsion preparation), stable during the FD process (Figure 1e,f and Appendix A), was then adopted as “baseline formulation” for following the studies. However, for pharmacokinetic profile and biodistribution studies, a more concentrated emulsion (estimated Fv = 17%) was used to inject the greatest number of droplets in a volume compatible with administration in mice. The latter concentrated emulsions also allow for an easier assessment of tissue biodistribution by ^19^F-NMR analysis. The preparation of each “baseline” and “concentrated” formulation is described in the experimental section. These two types of unlabeled emulsions were used to study the stability, biocompatibility and safety of droplets in vitro and in vivo. 

To allow the droplet follow-up after cellular internalization (in vitro) and BBB crossing (in vivo), a labeling of the previous baseline formulation was carried out. Two types of “labeled emulsions” were prepared according to the spectroscopic constraints of biological assays. DTE containing droplets simultaneously labeled on their core and surface was prepared to ensure imaging of the entire droplet. For its preparation, the surface of droplets was endowed with FITC moieties grafted to the surfactant polar head (the FITC-labeled surfactant constituted 10% of the droplet shell), while a hydrophobic dye (DiD), solubilized in the oily phase, was encapsulated in the core. As DiD absorbs at the DLS laser wavelength, size measurements of DTE were only performed by TEM (Appendix A). DTE was both used to study the cellular uptake and intracellular localization of droplets in vitro. In parallel, to avoid spectroscopic overlapping between fluorescein moieties and autofluorescence of tissues, an STE, labeled only with DiD in the droplet core, was prepared and used to follow droplets in vivo after BBB opening.

### 3.2. Emulsion Stability

#### 3.2.1. Droplet Dh Variation over Time 

To study the baseline emulsion stability over time, the droplet Dh was measured by DLS over 24 h at 4 °C (storage temperature) and at 37 °C (physiological temperature) after rehydration of freeze-dried emulsion with different volumes (150 or 300 µL) of various dispersing phases (water, NaCl 0.9% and complete cell culture medium (i.e., EMEM supplemented with FCS) to imitate physiological conditions). Dh changes did not seem to be related to the nature of the dispersing phase nor to the rehydration volume, but mainly to the storage temperature since, regardless of the nature of the dispersing phase, samples stored at 4 °C were more stable than those stored at 37 °C (Figure 2a,b). Indeed, for the same dispersing phase (NaCl 0.9%) and volume of dispersion (300 µL), the Dh of samples stored at 4 °C and 37 °C increased from 69 ± 1 nm to 102 ± 1 nm and from 68 ± 2 nm to 155 ± 2 nm over 24 h, respectively. To delay the expected emulsion ripening over time, for all in vitro and in vivo studies, freeze-dried formulations were rehydrated extemporaneously. Moreover, contrary to what was reported by Xiang et al. [56], sample dilution did not influence the Dh changes over time in our case, at either 4 °C or 37 °C (Figure 2c,d). Regardless of the conditions, during the first hour after rehydration, droplet size variation was limited over time to less than 13% growth, even at 37 °C, with Dh increasing from 69 ± 1 nm to 78 ± 2 nm, suggesting that droplets should not increase in size to more than 100 nm when diluted in the bloodstream.

#### 3.2.2. Passive Content Release

To study passive content release, an STE sample was used and stored at 37 °C. The dye (DiD) leakage from the core of droplets, assessed by HPLC, was less than 20% during the first 8 h, corresponding to the maximum duration of BBB-crossing after FUS-mediated BBB disruption for droplets with a size around 60 nm (Figure 2e) [21]. 

It is extremely important that the NC growth and content (drug or dye) release remain limited within the first few hours after injection to successfully pass through the permeabilized BBB and ensure an efficient brain delivery. Indeed, it is known that an increase in NC size is correlated to an easier uptake of particles by the reticuloendothelial system, leading to a decrease in their blood half-life [12,57]. Furthermore, there is a very limited probability that NCs with a diameter over 200 nm will cross the BBB after its FUS-mediated disruption. As a consequence, we can expect a good brain accumulation with our PFC nanodroplets whose size is lower than 100 nm, but large enough (>single nm particles) to bypass the glomerular filtration of the renal system [38,39]. 

Altogether, these results suggest that our PFC-emulsions have suitable characteristics for a future pharmaceutical development and biomedical use: optimal size of droplets, drug or dye loading by introduction of an oily phase, freeze-dried formulations for long term storage, stability over time, including in physiological conditions, and no burst release. We, therefore, continued our study with their in vitro and in vivo evaluation.

### 3.3. In Vitro Biocompatibility and Uptake of Nanodroplets

#### 3.3.1. In Vitro Biocompatibility 

Unsurprisingly, as shown in Appendix A, unloaded nanodroplets and possible derived compounds were biocompatible and safe in cancer and non-cancerous cell lines. Indeed, in vitro cell survival assays (assessed by using a colorimetric assay [33]) of both droplet components taken separately (surfactant F_8_TAC_13_ and ATBC) and of the whole baseline emulsion (expressed as surfactant concentration) were performed on one cancer cell line (U87-MG) and two non-cancerous cell lines (HMEC-1 and C8-D1A). For ATBC oil, no major reduction in cell viability was observed up to a concentration of 10 mg L^−1^, regardless of the cell line used. Regarding the surfactant (F_8_TAC_13_), the highest non-toxic concentration was 0.1 g L^−1^ for U87-MG and HMEC-1 cells. This result was consistent with the cell viability profile of droplets in the whole emulsion, which also showed cell viability higher than 80% for concentrations up to 0.1 g L^−1^ (expressed as surfactant concentration) for all of the three cell lines. It should also be noted that glioblastoma cells seemed to be more sensitive to droplets than non-cancerous cell lines, as their survival was distinctly lowered (35% ± 2%) for a droplet concentration (expressed as surfactant concentration) of 1 g L^−1^, and the difference was maintained for higher concentrations.

#### 3.3.2. In Vitro Uptake 

Along with their safety, it is necessary to ensure the potential of droplets as NCs for hydrophobic molecules. Therefore, a qualitative study of DTE cellular uptake was conducted with the U87-MG-transfected DsRed cell line using CLSM. Cells were exposed to DTE for 2, 4, 8 or 24 h. While absent in images of untreated cells, both fluorescence signals are observed in cells from 2 h of incubation with DTE (data not shown). The overlay of DsRed-transfected glioblastoma cells (red) with the two fluorescent probes (FITC, green and DiD, blue) strongly suggests the cellular uptake of double-tagged droplets (Figure 3a). The localization of nanodroplets inside the cytoplasm of cells was demonstrated by obtaining a Z-stack of 20 cross-sections along the thickness of the cell (data not shown), and after 24 h of incubation, the droplets still seem to be mainly located in the cytoplasm without penetrating the nucleus (Figure 3a). Moreover, the appearance of spherical green-blue dots suggests an undamaged droplet structure (Figure 3a) and supports the stability of nanodroplets in complete culture medium at 37 °C, as mentioned above.

Given the weak passive leakage of the droplet content, DTE can be used to study the cell internalization kinetics by FACS assay. Three different cell lines (U87-MG, HMEC-1 and C8-D1A cells) were each incubated with DTE for 1, 2, 4, 10 or 24 h. The number of fluorescence-positive cells counted per 10,000 cells, expressed as percentage of positive cells, and the fluorescence intensity of 10,000 treated cells normalized by that of untreated cells, expressed as their MFI, were collected from FACS data. Detectable MFI signals of both probes from cells only 1 h after incubation demonstrated a rapid uptake of double-tagged droplets, regardless of the cell line. Extending the incubation period was correlated with an increase in the percentage of doubly positive cells, suggesting a time-dependent kinetics. Glioblastoma cells (U87-MG) showed faster kinetics of internalization than non-cancerous cell lines (HMEC-1 and C8-D1A), with 90.8% ± 0.2% of doubly positive cells against 49.3% ± 0.7% and 43.5% ± 1.2% after 4 h of incubation, respectively (Figure 3b). This result is not surprising given that cancer cells and non-cancerous cells differ in various ways, dividing at an unregulated pace and having different cellular uptake pathways [58]. In addition, although more than 80% of cells were doubly positive after 10 h in all cell lines, MFI signals of the two fluorescent probes continued to increase significantly up to 24 h, suggesting an unsaturable internalization mechanism. Glioblastoma cells also showed a more important internalization of droplets than non-cancerous cells, with a 2.3-fold and 2.9-fold increase in FITC MFI signal, respectively, and a 2.3-fold and 6.7-fold increase in DiD MFI signal, respectively, after 24 h of incubation (Figure 3c,d). This can be correlated to the higher sensitivity to droplets of glioblastoma cells compared to non-cancerous cells described above. For each cell line, linear regression of MFI versus incubation time can suggest a steady rate of droplet internalization within the time interval tested, as reported by Nguyen et al. for magnetic NPs (Figure 3c,d) [59]. Herein, we demonstrate that the nanodroplets developed in this work are biocompatible and appear to be promising NCs, capable of delivering hydrophobic molecules into cells, including drugs, with probably faster uptake in cancer cells. The study was pursued with their in vivo evaluation in mice.

### 3.4. In Vivo Pharmacokinetics and Tissue Distribution of Nanodroplets

To assess the pharmacokinetic profile and tissue distribution of droplets, 13 groups of 5 female C57BL/6 mice were treated with a single dose of concentrated emulsion (300 µL of emulsion, i.e., a mean PFOB injection of 39.6 µL per mouse) by retro-orbital i.v. injection. PFOB concentration was quantified in blood and tissue samples using ^19^F-NMR.

#### 3.4.1. In Vivo Pharmacokinetics 

For pharmacokinetics, whole blood was collected 5, 10, 15 min, 1, 2, 4, 8, 16, 24, 48, 72 h, 7 and 14 days after injection. Among the 63 measurements of PFOB in blood, 21 records (33%) were below the Lower Limit of Quantification (LLOQ) and were excluded from the pharmacokinetic analysis (one mouse at the 24 h sampling point and all mice at the 48 h to 14 days sampling points). The retained data are summarized in Appendix A. The median PFOB concentration–time profile, overlaid with observed data points, is displayed in Figure 4a (same time points under the LLOQ were excluded). The pharmacokinetic curve indicates that droplets had a circulation time consistent with an efficient BBB crossing after FUS-mediated BBB disruption. The peak plasma concentration of PFOB (C_max_ = 25.90 µL mL^−1^ blood) was reached 2 h post-injection and could be explained by a possible persistence of droplets in the sinus after their retro-orbital injection. Eight hours after injection, a concentration of PFOB of 20.42 µL mL^−1^ blood remained in circulation corresponding to 79% of PFOB C_max_ (Figure 4a). The blood level of PFOB was undetectable (records below the LLOQ) 48 h post-injection. The linear terminal phase, observed on the semi-logarithmic scale (Appendix A), suggests a first-order elimination process (associated with a single-compartment model). Thanks to a non-compartmental analysis (a naive pooled data approach), mean pharmacokinetic parameters were calculated and are summarized in Table 2. Droplets were cleared from the mice body with a blood terminal half-life (t_1/2_), estimated to be around 3 h, opening up the feasibility of repeated administrations. The pharmacokinetic profile of PFOB emulsions reported in the literature appears to be similar, with a blood t_1/2_ ranging from 4 to 10 h following i.v. injection in rats [60]. Keipert et al. identified that PFOB blood t_1/2_ depends on the size of particles constituting the emulsion. For PFOB droplets with a diameter of the same order of magnitude as ours, mean blood t_1/2_ was 8 h [61]. The volume of distribution (V_d_), which reflects the degree of distribution of droplets in organs and tissues, was 0.93 mL, close to blood volume in the mice (1.2–2 mL for mice weighing between 16 g and 20 g), which suggests that droplets tend to remain in the systemic circulation and do not distribute easily to tissues. This observation allows us to foresee a reduced rate of peripheral adverse effects in the case of drug-loaded droplets.

#### 3.4.2. In Vivo Biodistribution 

To determine the distribution of droplets in tissues, major organs (liver, spleen, kidneys, brain, lungs and pancreas) were harvested at different times after droplet injection: 1, 8, 24, 48, 72 h, 7 and 14 days. The accumulation of PFOB was maximal in the liver and spleen 24 h after injection, with a maximum concentration of 14.3 ± 1.8 µL g^−1^ tissue and 6.6 ± 2.8 µL g^−1^ tissue, respectively. However, droplets were quickly eliminated from these two organs since PFOB concentrations dropped to 3.1 ± 1.0 µL g^−1^ tissue and 0.8 ± 0.2 µL g^−1^ tissue, 72 h after injection, respectively (Figure 4b), and were undetectable 7- and 14-days post-administration. This is consistent with the literature that describes an initial uptake of PFOB emulsion into the reticuloendothelial system and then an excretion by exhalation [62]. In other organs, PFOB concentrations were much lower with less than 1 µL g^−1^ tissue after 24 h (Figure 4b). The brain was the organ with the lowest PFOB concentration with a maximum of 0.06 ± 0.06 µL g^−1^, 8 h post-injection (Figure 4b).

Overall, these results indicate that the droplets remained in the bloodstream for several hours, with transient accumulation in some organs (mainly in the liver), and rapid elimination after 24 h, making it possible to consider repeated administrations. Furthermore, we did not observe any spontaneous intracerebral accumulation of droplets in animals with an intact BBB. Therefore, this PFC-emulsion associated with a BBB disruption technology appears to be an excellent NC for future applications in CNS disorders.

### 3.5. In Vivo Intracerebral Accumulation of Nanodroplets after BBB Disruption

Due to the high boiling point of PFOB, PFOB nanodroplets are not suitable to induce BBB permeabilization. In this study, the nanodroplets were only used as drug delivery systems. Therefore, FUS-mediated BBB disruption was performed in combination with SonoVue^®^, commercial MB usually used as ultrasound contrast agent. This technology has been demonstrated as a promising approach to induce BBB permeabilization for delivery of large pharmaceutic agents into the brain [20]. To confirm the efficiency of our FUS-mediated BBB disruption protocol, one mouse was first injected intravenously with SonoVue^®^ MB, then its right hemisphere was exposed to FUS and finally, Evans blue was intravenously administered. In the bloodstream, Evans blue binds to albumin, and the resulting Evans-blue-tagged albumin is extravasated into the brain parenchyma, only when and where the BBB is permeabilized. As shown in Figure 5a, blue coloration was observed only in the right hemisphere, indicating a successful BBB disruption of sonicated hemisphere after exposure to 0.33 MPa acoustic pressure (peak negative pressure estimated in situ).

To assess the delivery of droplets into the brain parenchyma after FUS-mediated BBB disruption protocol, STE was administered to mice 10 min before SonoVue^®^ MB injection and FUS application. We decided to disrupt the BBB after droplets injection to optimize their brain accumulation. Indeed, since they have a rather large size (tens of nm) and a rather long plasma half-life, it is preferable to permeabilize BBB when the droplets are already circulating, to take full advantage of the permeabilization time slot (maximum duration of BBB crossing for droplets with a diameter around 60 nm has been described to be 8 h) [21]. After BBB opening, mice were sacrificed, and undelivered droplets were washed from the harvesting. Brain tissue penetration of DiD-labeled droplets was first observed in four mice 1 h after BBB opening, after brain sectioning and using fluorescence microscopy. Delivery of droplets across the BBB was successful in all mice (*n* = 4). As can be seen from coronal brain slices (Figure 5b), DiD fluorescence signal from STE was clearly observed only in the hemisphere subjected to BBB disruption. By contrast, no fluorescence signal was observed in the hemisphere with non-disrupted BBB. Some assays were performed to assess the potential extravasation of STE after i.v. injection and FUS exposure without SonoVue^®^ injection, i.e., without BBB disruption. Interestingly, no DiD fluorescent signal from STE was observed on the brain (data not shown), suggesting that there is no DiD extravasation under these US conditions. We can, thus, assume that the injected DiD-loaded droplets were able to penetrate the brain tissue only after BBB disruption. The extent of droplet penetration was estimated by a semi-relative quantification of DiD fluorescence signal (DiD MFI) in sonicated and contralateral control hemispheres in these four mice. The mean percentage of DiD MFI increase in the BBB-permeabilized hemisphere was 6.6% ± 2.2% (*n* = 4 mice). Furthermore, the enhanced droplet accumulation in the sonicated hemisphere was also confirmed by FACS in three other mice 4 h after BBB opening (Figure 5c). For each mouse, the mean percentage of DiD-positive cells in the FUS-exposed hemisphere was significantly 8-fold, 2.8-fold and 2.1-fold higher than that of the contralateral control hemisphere, respectively (*p* < 0.05, *n* = 3 areas for each hemisphere). Increasing fluorescence intensity and DiD-positive cells in the BBB-disrupted hemisphere confirmed an efficient brain penetration of droplets using FUS combined with SonoVue^®^ MB. To our knowledge, this PFC-emulsion is one of the rare formulations that allows an increase in intracerebral fluorophore concentrations after BBB disruption [63]. Furthermore, FUS-mediated BBB disruption technology allows a localized delivery of our NC to a selected area of interest. In contrast, other brain delivery strategies, such as intranasal administration, enable BBB bypass without selectivity for brain regions of interest, leading to an increased risk of neurological adverse effects [64].

To further detect any microhemorrhage or tissue-damaging effects induced by droplets combined with the FUS procedure with Sonovue^®^ MB, histologic examinations with hematoxylin–eosin staining were performed after 24 h on coronal brain slices of one untreated mouse (control) and one mouse treated intravenously with STE. For each mouse, only the right hemisphere was subjected to FUS-mediated BBB disruption. No macroscopic brain damage or extravasation of erythrocytes around blood vessels could be observed for both mice (Figure 5d), suggesting a safe delivery of droplets into the brain after the FUS procedure. Furthermore, hemorrhages could also be detected by autofluorescence of blood using Green Fluorescent Protein channel of fluorescence microscopy. No petechia or hemorrhage were detected on brain slices with this method (Figure 5b). Contrary to other studies that showed brain accumulation of fluorescent NPs but with erythrocyte extravasations [39,65], we presented here a safe BBB permeabilization procedure leading to the efficient brain delivery of droplets. 

Taken together, these results imply that i.v. injection of PFC-emulsions in combination with FUS-mediated BBB disruption using SonoVue^®^ MB has the potential to achieve non-invasive, safe and localized drug delivery into the brain. In order to improve the BBB penetration and brain delivery of NCs, the use of brain-targeting ligands (i.e., peptides, antibodies…) shows high efficiency. However, the resulting targeted NCs are complicated to characterize, which is an obstacle to the translation of nanotechnology “from the bench to the clinic” [66]. Indeed, the characterization of nanoparticles for biomedical applications is a crucial step to obtain validation from health authorities and to allow their use in clinics [10,67]. Thus, in contrast to these complex tools, our easily characterized, long-term storable and “easy-to-use” PFC-emulsions, combined with FUS-mediated BBB disruption, constitute promising NCs for brain delivery and the treatment of CNS diseases.

## 4. Conclusions

In summary, we developed a novel and safe platform for the delivery of hydrophobic drugs into the brain after BBB disruption. FUS-mediated BBB disruption using SonoVue^®^ MB is a drug delivery method of high potential, currently showing promising results in clinical trials [16,17,68,69]. The characterization of our promising brain NC highlighted several properties fundamental to its clinical translational success: optimal nano size for biomedical use, ability to encapsulate hydrophobic agents, stability to FD allowing long-term storage, stability over time, including under physiological conditions, cell internalization, biocompatibility and tolerance after i.v. injection, long circulation time with no apparent toxicity and, finally, safe brain delivery after FUS-mediated BBB disruption. These emulsions appear as promising NCs for the future development of new pharmaceutical formulations for the treatment of brain diseases. To this end, hydrophobic drugs will be loaded into these nanodroplets and evaluated in vivo. In the future, the PFC core could benefit from other advantages, such as their use for imaging (^19^F-MRI or ultrasonography after the phase transition to bubbles) or as a sonosensitive system, to achieve the spatiotemporal control of drug release to desired sites (e.g., tumors).

## Data Availability

Not applicable.

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
