# Peer review of "Perfluorocarbon Nanodroplets as Potential Nanocarriers for Brain Delivery Assisted by Focused Ultrasound-Mediated Blood–Brain Barrier Disruption"

_pharmaceutics, 2022, doi:10.3390/pharmaceutics14071498_

Round 1

Reviewer 1 Report

Comments: The topic is interesting. However there are few  issues should to be addressed 

1-    There are many English typos errors should have been revised strongly over all the text.

2-    Authors didn't describe the mechanism by which nanodroplets can move in blood stream for long time circulation with no recognition by immune system. Authors should to describe this approach if there was no any functionalization for their surface.

3-    Scheme for chemical structure  illustrates  the possible attachment should be provided

4-    The mechanism by which DiD was being incorporated into core of nanodroplets was not written. it was not clear if authors used active technical method or passive technical method to integrate the dye into core and How can confirm their results.

5-    the accumulated DiD release as a model for encapsulation should to be extended into 72h and was being studied in physiological pH 7.4 and microenvironment of tumor pH 6.5.

6-    DiD  was used as a model . However, it is lipopholic molecules and many chemotherapies are hydrophobic. Would author explain  how their possible reaction and loading capacity can be performed?

7-    Glioblastoma cells  demonstrated  higher absorption for nanodroplets compared to non-cancerous cells. However, the mechanism of cellular internalization is still not clear. Mechanisms of phagocytosis, micropinocytosis, or endocytosis should have been studied.

8-    In line “364 “ section “histologic analysis” . Authors should please to change it into “ histological evaluation”.

9-    It is clear that focused ultrasound  allows nanodroplets to cross BBB. Authors should please to write “FUS-mediated BBB disruption” on the title.

10- It is still not clear if nanodroplets can further be adsorbed by other organs through ultrasound waves.

11- List of abbreviation should to be added. 

Author Response

Reviewer 1

English language and style

( ) Extensive editing of English language and style required
(x) Moderate English changes required
( ) English language and style are fine/minor spell check required
( ) I don't feel qualified to judge about the English language and style

Yes

Can be improved

Must be improved

Not applicable

Does the introduction provide sufficient background and include all relevant references?

( )

(x)

( )

( )

Are all the cited references relevant to the research?

( )

(x)

( )

( )

Is the research design appropriate?

( )

(x)

( )

( )

Are the methods adequately described?

( )

(x)

( )

( )

Are the results clearly presented?

( )

(x)

( )

( )

Are the conclusions supported by the results?

( )

(x)

( )

( )

Comments and Suggestions for Authors

Comments: The topic is interesting. However there are few  issues should to be addressed 

Responses to the reviewer (in blue)

Thank you for your reviewing and giving us the possibility to improve our manuscript in the best possible way. All new changes made in the manuscript have been marked up using the “Track Changes” function of MS Word.

  • There are many English typos errors should have been revised strongly over all the text. A correction of the English was done. We hope there are no remaining errors in the revised manuscript.
  • Authors didn't describe the mechanism by which nanodroplets can move in blood stream for long time circulation with no recognition by immune system. Authors should to describe this approach if there was no any functionalization for their surface.

Answer. We thank the reviewer for this relevant comment. Indeed, this mechanism is not described in the manuscript because we did not perform immunogenicity studies on the nanodroplets. What we know is that nanodroplets are coated at their surface with a nonionic surfactant which is non immunogenic by itself. In addition to their very low mean diameter (less than 150 nm over 24h, as shown on Figure 2, section 3.2.2.“Passive content release” pages 15-16) this feature probably confers “stealth” properties to the nanodroplets resulting in a long half-life (t1/2= 3.11h) in the bloodstream as reported in the “in vivo Pharmacokinetics” study on section 3.4.1. page 19.

  • Scheme for chemical structure  illustrates  the possible attachment should be provided

Answer. We may not understand the question. If it refers to the possible attachment of immune cells, as stated in answer 2, we assume that our nanodroplets are not immunogenic.

  • The mechanism by which DiD was being incorporated into core of nanodroplets was not written. it was not clear if authors used active technical method or passive technical method to integrate the dye into core and How can confirm their results.

      Answer. The DiD encapsulation into the core of droplets was performed by a “simple” dye solubilization into the oil phase (ATBC oil). The description of the preparation of DiD-loaded nanodroplets was improved in the revised manuscript for better understanding (see page 5 of revised manuscript). Briefly, after preparation of DiD-loaded nanodroplets through insonification using a 13 mm sonotrode, the un-encapsulated dye was separated from DiD-loaded droplets by several centrifugation-resuspension steps (as reported on section 2.6. Preparation of tagged “baseline” emulsions page 5). Moreover, after the last workup, a HPLC-analysis performed on the discarded supernatant confirmed that the concentration of residual un-encapsulated DiD was negligeable.

  • the accumulated DiD release as a model for encapsulation should to be extended into 72h and was being studied in physiological pH 7.4 and microenvironment of tumor pH 6.5.

Answer. The objective of the passive DiD release evaluation was to confirm the absence of burst release of DiD when droplets are injected in the bloodstream, so it was performed at short times (< 24h). Moreover, the estimated blood terminal half-life of droplets being around 3 hours and droplets being practically totally eliminated from the bloodstream at 24h (see Figure 4, section 3.4.1, page 21), our choice to evaluate the passive DiD release up to 24h was justified. Concerning the effect of pH, the objective of this work was the characterization of our nanocarrier for various medical applications (not only for cancer therapy), so stability was studied only in physiological conditions (37°C, pH 7.4, deionized water). Nevertheless, we agree with the reviewer that it might be interesting to study the behavior of our droplets at pH 6.5 when they will be loaded with an anticancer drug.

  • DiD  was used as a model . However, it is lipopholic molecules and many chemotherapies are hydrophobic. Would author explain  how their possible reaction and loading capacity can be performed?

Answer. DiD is a hydrophobic molecule (very low solubility in water) and, as such, it is lipophilic too. Many drugs are hydrophobic and therefore DiD is often chosen as a “model” hydrophobic molecule for drug carriers. The loading capacity is not the result of a “reaction” (if we understand the question correctly), but strongly depends on the dye solubility in oil. It could be “improved” by changing the oil nature. However, we must also balance the solubility of oil in water. In our case, ATBC gave the best trade-off between dye loading and dye release from droplets.

  • Glioblastoma cells  demonstrated  higher absorption for nanodroplets compared to non-cancerous cells. However, the mechanism of cellular internalization is still not clear. Mechanisms of phagocytosis, micropinocytosis, or endocytosis should have been studied.

Answer. Cancer cells and non-cancerous cells differ in various ways, dividing at an unregulated pace, and having different cellular uptake pathways or different internal membrane trafficking, as discussed in section 3.3.2 “In vitro uptake”. Concerning the mechanism of cellular internalization, it has been described that negatively or neutrally charged NPs with a size of a few hundred nanometers were mainly internalized via caveolin- and/or clathrin-mediated endocytosis (Behzadi, Shahed et al. “Cellular uptake of nanoparticles: journey inside the cell.” Chemical Society reviews vol. 46,14 (2017): 4218-4244; Foroozandeh, P., Aziz, A.A. Insight into Cellular Uptake and Intracellular Trafficking of Nanoparticles. Nanoscale Res Lett 13, 339 (2018).)

Moreover, it has been suggested that various cell types may employ different endocytotic pathways to internalize the same nanoparticle.

For these reasons, we assumed that the most likely internalization pathway of our nanodroplets is endocytosis, and did not investigate their exact internalization mechanism since it is cell-dependent and would not have been extended to other cell types.

  • In line “364 “ section “histologic analysis” . Authors should please to change it into “ histological evaluation”.

Answer. The change has been made.

  • It is clear that focused ultrasound  allows nanodroplets to cross BBB. Authors should please to write “FUS-mediated BBB disruption” on the title.

Answer. Title has been changed for “Perfluorocarbon nanodroplets as potential nanocarriers for brain delivery assisted by focused ultrasound-mediated blood-brain barrier disruption”.

  • It is still not clear if nanodroplets can further be adsorbed by other organs through ultrasound waves.

Answer. In this work, ultrasound waves are only focused on the brain, while the other organs are not exposed to ultrasound. Therefore, it seems clear that nanodroplets cannot be adsorbed (through ultrasound) by any organ except the brain thanks to the FUS-mediated BBB disruption. However, although not studied in our work, it is known that the use of ultrasound associated with the injection of microbubbles increases vascular permeability and allows drug delivery, particularly in tumors (Omata, Daiki et al. “Enhanced Vascular Permeability by Microbubbles and Ultrasound in Drug Delivery.” Biological & pharmaceutical bulletin vol. 44,10 (2021): 1391-1398). Thus, we could consider extending this system to other organs.

  • List of abbreviation should to be added.

Answer. A list of abbreviation has been added to the manuscript before “References” section. We let the editor assess whether this is appropriate.

Reviewer 2 Report

1.  Scheme 1, look like a graphical abstract, if it is allowed, It should be represented as that. 

2. Please cite methods (Emulsion stability determination, Cell culture, Confocal laser scanning microscopy (CLSM), Fluorescence-activated cell sorting (FACS),In vivo studies In vivo pharmacokinetics and tissue distribution, 19F-NMR PFOB quantitative determination in mice blood and organs, Intracerebral accumulation of STE using fluorescence microscopy, Intracerebral accumulation of STE using fluorescence microscopy etc. )  if not innovated for this study, 

3. Materials should be in manuscript in place of supp. document. 

Author Response

Reviewer 2

English language and style

( ) Extensive editing of English language and style required
( ) Moderate English changes required
( ) English language and style are fine/minor spell check required
(x) I don't feel qualified to judge about the English language and style

Yes

Can be improved

Must be improved

Not applicable

Does the introduction provide sufficient background and include all relevant references?

(x)

( )

( )

( )

Are all the cited references relevant to the research?

(x)

( )

( )

( )

Is the research design appropriate?

(x)

( )

( )

( )

Are the methods adequately described?

(x)

( )

( )

( )

Are the results clearly presented?

(x)

( )

( )

( )

Are the conclusions supported by the results?

(x)

( )

( )

( )

Comments and Suggestions for Authors

Responses to the reviewer (in blue)

We would like to thank you for reviewing our work. All new changes made in the manuscript have been marked up using the “Track Changes” function of MS Word.

  1. Scheme 1, look like a graphical abstract, if it is allowed, It should be represented as that.

Answer. We apology for this confusion !! Scheme 1 is not the graphical abstract. Indeed, we omitted to include our graphical abstract during the first submission. We now join the graphical abstract with the revised manuscript submission.

  1. Please cite methods (Emulsion stability determination, Cell culture, Confocal laser scanning microscopy (CLSM), Fluorescence-activated cell sorting (FACS),In vivo studies In vivo pharmacokinetics and tissue distribution, 19F-NMR PFOB quantitative determination in mice blood and organs, Intracerebral accumulation of STE using fluorescence microscopy, Intracerebral accumulation of STE using fluorescence microscopy etc. )  if not innovated for this study,

Answer. This has been done in the revised manuscript when relevant. However, most of the described methods have been specifically tailored for this study.

  1. Materials should be in manuscript in place of supp. document.

Answer. All materials and methods have been integrated into the manuscript.

Reviewer 3 Report

The combination of gas-filled micro-bubbles and focused ultrasonund allows to reversibly permeate the BBB.  By applying this procedure  a safe method for delivery of hydrophobic drugs encapsulated in perfluorocarbon nanodroplets into the brain has been demonstrated. PFOB is the core of the nanodroplets The droplets stabilization and dispersability in water was pursued by anchoring a fluorinated surfactant at their surfaces. The reported PFC-emulsions appear to be suitable for a clinical traslaction (optimal size, good ability to uptake athe payload in the oily phase, good stability properties,...).

The experiments were carefully planned and executed and the results well interpreted. It is a nice, thorough work. I reccomend publication as it stands.

Author Response

Reviewer 3

English language and style

( ) Extensive editing of English language and style required
( ) Moderate English changes required
(x) English language and style are fine/minor spell check required
( ) I don't feel qualified to judge about the English language and style

Yes

Can be improved

Must be improved

Not applicable

Does the introduction provide sufficient background and include all relevant references?

(x)

( )

( )

( )

Are all the cited references relevant to the research?

(x)

( )

( )

( )

Is the research design appropriate?

(x)

( )

( )

( )

Are the methods adequately described?

(x)

( )

( )

( )

Are the results clearly presented?

(x)

( )

( )

( )

Are the conclusions supported by the results?

(x)

( )

( )

( )

Comments and Suggestions for Authors

The combination of gas-filled micro-bubbles and focused ultrasonund allows to reversibly permeate the BBB.  By applying this procedure  a safe method for delivery of hydrophobic drugs encapsulated in perfluorocarbon nanodroplets into the brain has been demonstrated. PFOB is the core of the nanodroplets The droplets stabilization and dispersability in water was pursued by anchoring a fluorinated surfactant at their surfaces. The reported PFC-emulsions appear to be suitable for a clinical traslaction (optimal size, good ability to uptake athe payload in the oily phase, good stability properties,...).

The experiments were carefully planned and executed and the results well interpreted. It is a nice, thorough work. I reccomend publication as it stands.

Responses to the reviewer (in blue)

Answer. Thank you very much for your appreciation of this work.

Reviewer 4 Report

Authors present a very interesting application of perfluorocarbon  nanodroplets as ultrasound-responsive nanocarriers to overcome BBB. I would highly recommend the article given its significance and quality presentation of the data. It can be said that the article is very well written while the results are supported by the experimental data. I would advise the follow comments to authors before publication

Comment 1. Please check for minor syntax and grammar errors

Comment 2. It would be better if you don't use abbreviations in the abstract

Comment 3. Add any references for the methodology you used and discuss the results with already published data

Comment 4.  You should report that the entrapment of the hydrophobic agents will be a future study

Author Response

Reviewer 4

English language and style

( ) Extensive editing of English language and style required
( ) Moderate English changes required
(x) English language and style are fine/minor spell check required
( ) I don't feel qualified to judge about the English language and style

Yes

Can be improved

Must be improved

Not applicable

Does the introduction provide sufficient background and include all relevant references?

(x)

( )

( )

( )

Are all the cited references relevant to the research?

(x)

( )

( )

( )

Is the research design appropriate?

(x)

( )

( )

( )

Are the methods adequately described?

(x)

( )

( )

( )

Are the results clearly presented?

(x)

( )

( )

( )

Are the conclusions supported by the results?

(x)

( )

( )

( )

Comments and Suggestions for Authors

Authors present a very interesting application of perfluorocarbon  nanodroplets as ultrasound-responsive nanocarriers to overcome BBB. I would highly recommend the article given its significance and quality presentation of the data. It can be said that the article is very well written while the results are supported by the experimental data. I would advise the follow comments to authors before publication

Responses to the reviewer (in blue)

We are very grateful for your reviews of our manuscript. Thank you for giving us the possibility to improve our manuscript in the best possible way. All new changes made in the manuscript have been marked up using the “Track Changes” function of MS Word.

Comment 1. Please check for minor syntax and grammar errors

Answer. A careful rereading of the manuscript has been carried out. We hope there are no remaining errors in the revised manuscript.

Comment 2. It would be better if you don't use abbreviations in the abstract

Answer. Abbreviations have been removed from the abstract.

Comment 3. Add any references for the methodology you used and discuss the results with already published data.

Answer. For references in the methodology, this has been done in the revised manuscript when relevant. However, most of the described methods were specifically tailored for this study. We also reinforced the discussion of our results with comparison to the state of the art.

Comment 4.  You should report that the entrapment of the hydrophobic agents will be a future study

Answer. We agree with the reviewer. In the conclusion, the sentence “To this end, hydrophobic drugs could be loaded into nanodroplets and evaluated in vivo.” was changed by “To this end, hydrophobic drugs will be loaded into these nanodroplets and evaluated in vivo”.

Round 2

Reviewer 1 Report

Manuscript was revised point by point according to reviewer comments and It is more acceptable NOW.